# Aestivation in Nature: Physiological Strategies and Evolutionary Adaptations in Hypometabolic States

**DOI:** 10.3390/ijms241814093

**Published:** 2023-09-14

**Authors:** Chunxi Jiang, Kenneth B. Storey, Hongsheng Yang, Lina Sun

**Affiliations:** 1CAS Key Laboratory of Marine Ecology and Environmental Sciences & Experimental Marine Biology, Institute of Oceanology, Chinese Academy of Sciences, Qingdao 266071, China; jiangchunxi@qdio.ac.cn (C.J.); hshyang@qdio.ac.cn (H.Y.); 2Laboratory for Marine Ecology and Environmental Science & Marine Biology and Biotechnology, Qingdao National Laboratory for Marine Science and Technology, Qingdao 266237, China; 3University of Chinese Academy of Sciences, Beijing 100049, China; 4Department of Biology, Carleton University, Ottawa, ON K1S 5B6, Canada; kenstorey@cunet.carleton.ca

**Keywords:** aestivation, hypometabolic states, physiological strategies, regulatory network

## Abstract

Aestivation is considered to be one of the “purest” hypometabolic states in nature, as it involves aerobic dormancy that can be induced and sustained without complex factors. Animals that undergo aestivation to protect themselves from environmental stressors such as high temperatures, droughts, and food shortages. However, this shift in body metabolism presents new challenges for survival, including oxidative stress upon awakening from aestivation, accumulation of toxic metabolites, changes in energy sources, adjustments to immune status, muscle atrophy due to prolonged immobility, and degeneration of internal organs due to prolonged food deprivation. In this review, we summarize the physiological and metabolic strategies, key regulatory factors, and networks utilized by aestivating animals to address the aforementioned components of aestivation. Furthermore, we present a comprehensive overview of the advancements made in aestivation research across major species, including amphibians, fish, reptiles, annelids, mollusks, and echinoderms, categorized according to their respective evolutionary positions. This approach offers a distinct perspective for comparative analysis, facilitating an understanding of the shared traits and unique features of aestivation across different groups of organisms.

## 1. Introduction of Animal Aestivation

### 1.1. Research Background

#### 1.1.1. Several Typical Hypometabolism Regulations

When faced with unfavorable environmental conditions, organisms adjust their ATP consumption by relying on their own fuel reserves and reducing their metabolic rate to prolong their survival and maintain viability. Depending on the species, periods of dormancy can last for days, months, or even years before arousing to an active state and resuming growth, development, and reproduction when environmental conditions become favorable again. This strategy is known as hypometabolic regulation. Dormancy and cryptobiosis can be roughly categorized as hypometabolic regulation. When animals are in a state of dormancy, they can still move and respond to external stimuli, but they are not very active and have a very low metabolic rate. The difference between the cryptobiotic phenomenon and dormancy is that the former usually refers to a state of complete lack of metabolism caused by changes in osmotic and hydration status under extreme environmental conditions.

Dormancy is a broad concept that encompasses survival strategies such as aestivation, hibernation, torpor, and diapause. These survival strategies share close relationships and similar regulatory mechanisms, making them ideal for comparative studies by researchers. The distinction between aestivation and hibernation is dependent on the season or environmental conditions when the animal enters a hypometabolic state. Aestivation is often employed to resist environmental conditions such as high temperatures and water shortages, whereas hibernation is used to resist low temperatures and freezing. The duration of aestivation or hibernation bouts depends on the cycle of changes in environmental conditions and the species tolerance range. 

Torpor typically refers to a daily pattern of dormancy that is observed primarily in small endotherms (birds and mammals). During the active part of the day, these animals maintain a euthermic (normal) body temperature and activity levels. However, during a portion of the day (usually at night), they allow their metabolic rate and body temperature to drop as a means of conserving energy. Diapause is a dormancy that is commonly observed in arthropods (e.g., insects, crustaceans, etc.) as well as various other invertebrates and vertebrates and consists of a pause in morphogenesis/growth and a decrease in physiological activity, often as a response to environmental conditions. Once initiated, organisms must pass through a certain developmental stage or come to an end after undergoing specific physiological changes. The main difference between diapause and other dormancy strategies is that diapause often occurs at a specific developmental stage, is relatively stable, and is commonly induced by certain seasonal signals before the onset of unfavorable conditions. 

#### 1.1.2. Physiological and Behavioral Characteristics of Aestivation

Aestivation refers to a reversible metabolic inhibition displayed by both homeotherms and endotherms when faced with environmental challenges such as high temperatures, drought, or a lack of food. However, the physiological and behavioral characteristics of aestivation vary across different species. For instance, in sea cucumbers (*Apostichopus japonicus*), aestivation lasting up to 100 days is marked by cessation of feeding, degenerative atrophy of the digestive tract, metabolic inhibition, and weight loss. Nevertheless, normal structure and function can be restored upon release from aestivation [1]. On the other hand, the African lungfish (*Protopterus* sp.) survives the dry summer months in an evaporating pond by secreting a mucus cocoon around itself and surviving for months or even years in a hypometabolic state. During this period, its immune system continues to function to protect the skin from pathogens [2]. African clawed frogs (*Xenopus laevis*) have a similar strategy, digging into and remaining immobile in the mud during the seasonal drying of their lakebed habitat. During this time, they can lose up to 30% of their total body water due to dehydration, resulting in increased blood viscosity and impaired oxygen transport. However, muscle and liver cells remain intact during dormancy and resume normal activity at the end of dormancy [3]. By contrast, various terrestrial vertebrates, such as milk snails (*Otala lactea*), are seasonally exposed to high temperatures and limited food and water. To resist these harsh conditions, snails can spend several months in aestivation, where their basal metabolic rate is strongly reduced to less than 30% of their normal resting rate. A mucus epiphragm is secreted over the shell’s aperature to aid in resisting desiccation. This allows snails to reallocate energy use to vital cellular processes and protect/stabilize macromolecules in the face of adverse environmental conditions, thereby maximizing survival [4].

#### 1.1.3. Comparison of Aestivation in Ectotherms and Endotherms

Aestivation is often induced by high summer temperatures, which raises the question of how aestivating animals regulate their body temperatures in response to environmental changes. Here, strategies of aestivation can be divided into two groups: ectotherms and endotherms.

Since their divergence from their ectothermic ancestors, endotherms have developed a variety of behavioral, morphological, physiological, and biochemical traits to regulate their endogenous metabolic heat production and maintain a consistent body temperature (Tb) despite variable ambient temperatures (Ta). This thermoregulatory mechanism is adjusted during aestivation or torpor, during which the thermogenic metabolism may undergo rapid changes in certain phases of aestivation or torpor, temporarily departing from Tb. Nonetheless, the theropod’s overall body temperature remains constant. The only way for endotherms to maintain their body temperature (Tb) within the thermoneutral zone (TNZ) is by generating more metabolic heat. The majority of metabolic heat is derived from the mitochondrial electron transport system (ETS), which accompanies the oxidative release of nutrients. A portion of the free energy produced by this system is released as heat, and therefore, endotherms provide an endogenous heat source for regulating Tb through their high metabolic rate.

The body temperature (Tb) of ectotherms is determined by the ambient temperature (Ta) to which they are exposed during dormancy. As a result, the metabolic rate (MR) of ectotherms decreases exponentially with decreasing Ta. Ectotherms and endotherms differ in terms of proton leak and metabolic energy release. Endotherms bind the ADP released from ATP hydrolysis to the central channel of the ETS complex V, thereby increasing the channel for protons to flow from the intermembrane space into the matrix. This proton flow not only drives ATP synthesis to maintain metabolic homeostasis but also leads to the restoration of proton motive force, releasing a significant amount of heat that can be used to regulate Tb. However, ectotherms reduce pathways of ATP consumption, including ion pumps and protein synthesis, as a key mechanism to maintain tissue metabolic balance. As a result, ectotherms appear to decrease proton leaks from mitochondria to conserve metabolic energy [5].

### 1.2. Main Elements of Animal Aestivation Research

#### 1.2.1. Antioxidant Defense

The antioxidant defense system is a well-conserved biochemical mechanism that safeguards organisms against the deleterious impact of metabolic byproducts of reactive oxygen species (ROS). Both invertebrates and vertebrates have developed multiple mechanisms to regulate metabolic pathways in response to environmental changes such as photoperiod, temperature, salinity, humidity, oxygen content, and food availability. The antioxidant defense system is one such mechanism [6,7]. Species that undergo aestivation often experience a temporary rise in oxygen consumption during the awakening process, resulting in an increased production of reactive oxygen species (ROS) and higher concentrations of redox-regulated molecules in relevant tissues. However, excessive ROS concentrations can be toxic to organisms, damaging cellular function by oxidizing biomolecules [8]. Therefore, organisms must enhance their antioxidant defenses before exposure to environmental or tissue hypoxia, an adaptation known as the “preparation for oxidative stress” (POS) strategy that was first identified in the 1990s [9,10,11].

The POS theory, proposed more than two decades ago, has been demonstrated in nearly 100 species. Its molecular mechanism involves metabolic inhibition during aestivation, which can lead to tissue hypoxia. Hypoxia-tolerant animals induce more reactive oxygen species (ROS) production under hypoxia than normoxia. The overproduction of mitochondrial ROS acts as a physiological regulatory signal that triggers the cellular signaling pathway of oxidative stress, activating redox-sensitive transcription factors such as HIF, Nrf2, NF-kB, and FoxO at the cellular level. In turn, this activation upregulates the expression of antioxidant defense enzymes (including CAT, SOD, GR, GSH-Px, GST, etc.) and molecular chaperones, such as HSP70 and HSP90. This process triggers changes in cellular phosphorylation status and microRNA regulation, preparing the organism for a massive increase in ROS and avoiding cellular damage [9,12,13].

#### 1.2.2. Nitrogen Metabolism and Ammonia Detoxification

Prolonged inactivity and a lack of food during aestivation can have a significant impact on an organism’s nitrogen metabolism. Recent studies have examined changes in nitrogen metabolism, conversion of toxic ammonia, and urea excretion during aestivation in three phases: induction into aestivation, maintenance of aestivation, and arousal from aestivation. Several studies have indicated that the synthesis and accumulation of urea, an internal signal during the induction phase, may be a crucial factor in the initiation and maintenance of aestivation [14,15]. During this period, there are significant changes in nitrogen metabolism. Ammonia production decreases, whereas urea synthesis increases. The rate of protein synthesis and turnover increases, but the rate of protein degradation remains unchanged. In the maintenance phase, protein synthesis generally decreases, except in specific organs such as the epithelial skin and gonadal tissue of desert frogs or the hepatopancreas and foot muscles of estivating snails, where proteins play an essential role in immunity or structural maintenance of tissues. Increased hepatic amination activity by hepatic GDH leads to suppressed ammonia production, whereas the increased activity of ornithine-urea cycle enzymes leads to higher rates of urea synthesis [16,17]. During the arousal period from aestivation, the organism’s absorption of water from the environment may act as a signal for arousal. Water absorption must occur before urea excretion because urea promotes the osmotic absorption of water. After water absorption, accumulated urea within the body begins to be excreted. This process is facilitated by an increase in urea transporter proteins or by an increase in the production of endogenous ammonia [18], which promotes urea excretion. The increase in urea excretion and changes in tissue fluid osmolality indirectly influence tissue regeneration after the arousal phase of aestivation. Specifically, a decrease in urea levels can result in an osmotic imbalance that triggers cell membrane rupture, which is immediately followed by the removal of damaged cells by macrophages, and then tissue regeneration begins [19].

In summary, during the induction and maintenance periods of aestivation, a decrease in ammonia production and an increase in urea synthesis are commonly observed, whereas during the arousal period, water absorption and urea excretion are observed [20]. With regard to urea synthesis, most hibernating species convert toxic ammonia into urea through the ornithine-urea cycle. Although this reduces the damage caused by toxic ammonia to the body, the accumulation of urea can also reach toxic levels. So why synthesize and accumulate urea instead of non-toxic nitrogenous waste such as purines or uric acid? There are several hypotheses: (1) Urea synthesized during the maintenance stage of hibernation needs to be excreted after arousal, and the energy required for urea excretion is lower than for purines. (2) Urea is not a nitrogenous waste product of hibernation metabolism but rather a nitrogen-containing product with specific functions. For example, the accumulation of urea can reversibly inhibit key metabolic enzymes, thereby reducing energy consumption. (3) An increase in urea levels changes the osmotic pressure, which helps absorb water from outside the body or retrieve water stored in the bladder during the late stages of hibernation [21]. (4) Urea accumulation can also lower the ionic strength of cells, which could otherwise have a negative effect on enzyme activity [19]. In other words, the accumulation of urea during hibernation is a balance between energy consumption, toxin accumulation, water retention, enzyme activity maintenance, and body function.

#### 1.2.3. Lipid Reserve and Lipid Metabolism

Lipid stores are comprised mainly of triglycerides, fat bodies, and droplets, with triglycerides being the most energy-dense form of energy storage. When nutrients are scarce, triglycerides are converted into free fatty acids and released into the circulation. Metabolically active tissues absorb and oxidize these fatty acids to fuel the production of ATP. Aestivating animals typically experience months or even years of starvation. Therefore, understanding how they balance lipid synthesis, accumulation, and catabolism in a low metabolic state and how they adjust substrate utilization from glucose oxidation to fatty acid oxidation according to their metabolic needs and substrate availability is an essential research question in aestivation [22].

Aestivating animals, such as Australian leptodactylid frogs, African lungfish, and land snails, have developed a survival strategy of increasing their fat reserves before aestivation and relying on these reserves to supply their metabolic fuel during aestivation [23,24,25]. However, they face the challenge of balancing fat accumulation and utilization. Aestivating animals appear to increase fat accumulation before aestivation through a mechanism of insulin resistance, which regulates the uptake and oxidation of glucose, promotes gluconeogenic processes in the liver, and inhibits systemic anabolism to conserve energy. Upon awakening, these animals rapidly become insulin-sensitive again. Additionally, aestivating animals should be capable of rapidly catabolizing and utilizing fats through the regulation of the oxidative breakdown process of fatty acids. Pyruvate dehydrogenase kinase isoenzyme 4 (PDK4) is thought to play a crucial role in the metabolic flexibility of dormancy by inhibiting the pyruvate dehydrogenase complex (PDC) via phosphorylation, thus controlling cellular fuel selection and ultimately favoring fatty acid oxidation. Furthermore, increasing the overall protein expression of key enzymes in the fatty acid catabolic pathway, including those involved in fatty acid transport, mitochondrial entry, β-oxidation, the TCA cycle, and the electron transport chain pathway, also promotes fatty acid oxidation [26,27]. Furthermore, ketone bodies, intermediate metabolites of fat oxidative metabolism, act as a substitute for glucose and become the main energy supplier for tissues with a high predilection for glucose, such as the heart and neurons [22]. Finally, these animals should also have the capacity to tolerate high levels of fat accumulation, as high-fat accumulation is generally considered to threaten the organism’s health. Research has shown that high-fat accumulation in dormant animals does not lead to immune system dysregulation because they invest less in their immune system during summer dormancy. This results in fewer immune cells being produced in response to the same pathogenic stimulus and reduced immune system sensitivity. However, this increases the risk of being victimized by pathogens or predators [26]. Further exploration is needed to understand how organisms balance their immune and metabolic needs during dormancy.

#### 1.2.4. Muscle Disuse Atrophy and Muscle Protection

Muscle damage and atrophy during aestivation result from three primary factors. First, when fat reserves are exhausted, protein becomes the primary fuel for maintaining basal metabolic processes, and the structure and function of muscle, as part of the protein composition, may be altered. Second, prolonged periods of inactivity during aestivation can cause disuse atrophy of certain muscles, as observed in various clinical trial models in mammals. Finally, high temperatures during aestivation result in increased oxygen consumption in ectothermic animals, leading to the overproduction of reactive oxygen species (ROS) that can cause oxidative stress, which in turn can lead to oxidative damage and disuse atrophy of muscles [28].

However, the maintenance of muscle morphology and function, especially in skeletal muscles, which are critical for locomotion, is essential for normal behaviors such as feeding and reproduction after aestivation. Therefore, aestivating animals have evolved various strategies to reduce disuse atrophy and maintain locomotor ability. Strategy 1 involves the independent function and regulation of different muscle tissues during aestivation. In frogs, smaller non-jumping muscles serve as a protein source during aestivation and are prioritized over larger jumping muscles [29]. Skeletal and cardiac muscles of aestivating frogs have significantly different rates of mitochondrial reactive oxygen species production, with skeletal muscle exhibiting substantial inhibition while cardiac muscle maintains normal production [30]. Strategy 2 involves avoiding atrophy by suppressing the metabolic rate. Aestivating frogs downregulate the expression of metabolism-related enzymes such as NADH ubiquinone oxidoreductase subunit 1 and ATP synthase and increase the level of post-translational dephosphorylation of related enzymes and other proteins, thereby significantly inhibiting metabolic rate and reducing muscle damage from reactive oxygen species (ROS) [31,32]. 

Strategy 3 involves limiting atrophy by regulating oxidative stress. Aestivating frogs enhance their defense against oxidative damage by increasing or maintaining endogenous antioxidants in their jumping muscles [29]. Additionally, the activities of antioxidant-related enzymes, such as catalase, glutathione peroxidase, uncoupling protein 2, glutathione-S-transferase, and glutathione reductase, are maintained during aestivation [31,33]. In African clawed frogs (*Xenopus laevis*), the Akt-FoxO signaling pathway is maintained during aestivation by decreasing Akt phosphorylation levels and activating the transcription factors FoxO1 and FoxO3 to regulate oxidative stress, thereby inhibiting muscle atrophy [34]. Strategy 4 involves limiting atrophy by reducing myostatin (mstn/Mstn) expression. Although African lungfish show a significant increase in mstn/Mstn expression levels early in aestivation, the expression levels are controlled as muscle degeneration progresses. By the time of aestivation arousal, myostatin levels have been reduced to control levels [35].

#### 1.2.5. Antibacterial Immune Protection

Hypometabolic strategies, including aestivation, help organisms survive unfavorable environmental conditions. These strategies also dramatically reduce the resistance of dormant organisms to predators and pathogens. Firstly, organisms typically need to accumulate large amounts of fat for energy supply during dormancy, and in order to avoid various disease responses triggered by high fat, they typically reduce investment in the immune system during dormancy [26]. Secondly, high environmental temperatures during aestivation significantly reduce the activity of metabolic and immunological enzymes. For example, the activities of total superoxide dismutase (T-SOD), catalase (CAT), myeloperoxidase (MPO), and lysozyme (LZM) are significantly reduced in aestivating sea cucumbers (*A. japonicus*), and the total body cavity cell count (TCC) is also negatively correlated with temperature, resulting in weaker cellular immune activity during aestivation [36]. Thirdly, mounting an immune response is an energetically expensive process, and a hypometabolic state with limited energy availability can impede immune function, as the synthesis of many new proteins, including cytokines, chemokines, and acute phase reactants, requires energy trade-offs [37].

However, antimicrobial immune protection during aestivation is essential for hypometabolic organisms that remain motionless for long periods, and they have evolved unique immune adaptations. One of the most common adaptations is the mucus callus of the African lungfish. To survive the dry summer months, African lungfish (*Protopterus* sp.) live in dry mucus cocoons (dormant cocoons) for months or even years. The formation of dormant cocoons is a defense strategy for this metabolically retarded vertebrate, as it isolates bacteria outside the body [38]. Recent studies have shown that the dormant cocoon of lungfish is a living tissue with a well-defined cellular structure rather than a dead, dry mucus layer. Lungfish form cocoons through the successive shedding of multiple regenerated, inflamed epidermal layers. This living cell cocoon actively transcribes immune genes and traps bacteria, providing long-term antimicrobial protection. During tissue remodeling, a pro-inflammatory microenvironment is created on the skin surface, which recruits granulocytes into the callus to release extracellular traps (ETs). The ETosis pathway plays a crucial role in infection, inflammation, injury, tissue remodeling, and autoimmunity, providing immune protection for these aestivation lungfish [2,39]. In addition, freshwater snails (*Pila globose*) demonstrate immune-resilient adaptation to different environmental and organismal changes by exhibiting significant changes in immune-related parameters in hemocytes and hemolymph at different stages of aestivation [40]. Similarly, hibernating bats adopt the intermittent arousal and local immunity strategy, allowing them to respond to local tissue pathogens during brief periods of body temperature recovery while balancing immunity against other energy expenditure during prolonged hibernation [41]. Other dormant animals in a hypometabolic state also exhibit unique immune strategies worth noting. For instance, hibernating thirteen-lined ground squirrels (*Ictidomys tridecemlineatus*) only experience a reduction in their ability to induce humoral immune responses to TI-2 antigens (T-cell non-dependent antigens), whereas their ability to induce humoral responses to TD antigens (T-cell-dependent antigens) is maintained [42].

#### 1.2.6. Visceral Degeneration and Regeneration

A number of studies have shown that estivating animals often experience degenerative atrophy of internal organs, particularly those of the digestive tract, after months or even years of fasting [22,43,44,45,46,47]. These include studies of the green-striped burrowing frog (*Cyclorana alboguttata*), the three-toed amphibian (*Amphiuma tridactylum*), sea cucumbers (*A. japonicus*), African lungfish (*Protopterus annectens*), and tegu lizards (*Salvator merianae*), suggesting a universality of this response. Interestingly, these animals are able to efficiently digest their first meal upon awakening from aestivation, suggesting a rapid recovery of gut morphology and function prior to (or coincident with) the start of re-feeding [22,43,44,47]. 

Visceral degeneration facilitates the coordination of metabolic conservation. There is significant variation in the rate of mass loss in different tissues and organs during aestivation. For example, adipose tissue is typically broken down for use in energy (ATP) production, whereas gastrointestinal organs (oesophagus, stomach, small intestine, and hindgut) degenerate due to prolonged non-use, resulting in a rate of cell loss that exceeds the rate of proliferation during aestivation. Organs such as the pancreas, liver, and gallbladder also lose mass due to a lack of neural stimulation activated by food intake. Conversely, some organs, including gills, skin, heart, and kidneys, retain much of their mass and structure during aestivation because they are essential for life activities such as respiration, protection, blood circulation, and nitrogenous waste removal [22]. Despite the significant reduction in gut function during aestivation, some cellular processes, such as L-proline transport rates, Na^+^/K^+^-ATPase activity, and protein synthesis, are maintained to maximize absorption efficiency during re-feeding. A continuous stimulation of the intestinal nutrient transport systems helps to maintain their function, ensuring that substrates that enter the intestinal lumen by passive diffusion or cell shedding are recycled by the intestine rather than lost [48]. In addition, many studies have described the molecular mechanisms of intestinal degeneration, cytoprotection, and hypometabolism during aestivation in sea cucumbers (*A. japonicus*). Intestinal degeneration is caused by apoptosis, and most apoptotic events occur within 20 days after an increase in ambient temperature. The expression of apoptotic genes such as Ajcaspase-8 (CASP8) and Ajcaspase-3 (CASP3) is elevated during this period [45]. However, anti-apoptotic regulation takes place to avoid excessive degeneration of the intestine by coordinating the relationship between heat shock protein 70 (HSP70) and apoptosis-inducing factor mitochondrial 1 (AIFM1), a bifunctional flavoprotein involved in the non-dependent activation of apoptotic caspases. During aestivation, the gene and protein expression profiles of AjAIFM1 and AjHSP70 were negatively correlated, with AjAIFM1 being repressed and AjHSP70 being strongly up-regulated. Furthermore, there is a potential interaction of the AjHSP70 protein with AjAIFM1, which may help to avoid excessive degradation of the intestine during aestivation [49]. Moreover, DNA methylation levels increased significantly in the intestine during aestivation, and the phenomenon of intestine hypometabolism was caused by transcriptional repression of various metabolic pathways mediated by DNA hypermethylation. Hypermethylated genes (HMG) are involved in a variety of metabolic pathways, such as carbon metabolism, fatty acid metabolism, pyruvate metabolism, and retinol metabolism [50]. 

Visceral regeneration facilitates adaptation to refeeding. The process of awakening from aestivation induces significant changes in the morphological parameters of the intestine. Gastrointestinal organs initiate regeneration to accommodate the digestion of food, leading to an increase in the masses of small and large intestines, small intestinal length, longitudinal fold (LF) height, enterocyte cross-sectional area, and microvilli height and density [43]. Recently, there have been increasing numbers of studies aimed at elaborating on the molecular mechanisms underlying intestinal regeneration. Among these studies, transcriptomic analyses of intestine regeneration in *A. japonicus* have revealed that differentially expressed genes are significantly enriched in the ribosome and spliceosome pathways, the Notch signaling pathway, ECM-receptor interaction, and cytokine-cytokine receptor interaction. Intestinal regeneration is a multi-gene regulated process involving multiple signaling pathways, such as Wnt, Hippo, FGF, and vincristine, and gene families including Fgfr, PSP94-like, fibrinogen-related protein, Wnt, Frizzled, Hox, and Mtf [50].

#### 1.2.7. Epigenetic Regulation during Aestivation

Aestivation is a common hypometabolic survival strategy where metabolic rate is reduced to 1–20% of the resting metabolic rate, and this inhibition is believed to be reversible. Research has demonstrated that aestivating animals can achieve reversible repression of gene expression in a hypometabolic state through epigenetic regulatory mechanisms, including DNA methylation, post-translational modifications of histones such as acetylation and phosphorylation, changes in miRNA expression, and preservation of mRNA transcripts [51,52].

##### DNA Methylation

During aestivation, DNA methylation plays a crucial role in transcriptional silencing. Methylation can impede transcription factor binding or recruit repressive proteins such as MBD1, MBD2, and MeCP2. Upregulation of DNA cytosine-5 methyltransferase 1 (DNMT1) and transcriptional co-repressor (SIN3A) occurs during aestivation. DNMT1 generates transcriptional repression modifications that are stable and reversible, while SIN3A is a co-repressor that promotes deacetylation, which reduces gene upregulation [51]. DNA methylation analysis of the aestivation intestine of *A. japonicus* revealed a significant increase in DNA methylation levels. Among the 411 hypermethylated genes (HMG) identified, 64% showed transcriptional repression during aestivation. These genes are involved in metabolic pathways such as carbon metabolism, fatty acid metabolism, pyruvate metabolism, and retinol metabolism [50]. Whole-genome methylation sequencing of aestivating *A. japonicus* showed that differentially methylated genes were enriched in the mRNA surveillance pathway, metabolic pathway, and RNA transport pathway. The mRNA surveillance pathway ensures mRNA quality, detects translation errors, and degrades abnormal mRNAs. It is necessary for oxidative stress tolerance, which inhibits the non-stop decay of abnormal proteins produced by translation errors. Potentially essential regulators for aestivation include many retrovirus-related transposon genes modified by DNA methylation [53].

##### Post-Translational Modifications of Histones

Acetylation could increase transcriptional activity, and therefore transcriptional repression during hypometabolism is often achieved by deacetylation in aestivation animals. Almost every enzyme in the metabolic pathways of aestivation, including glycolysis, gluconeogenesis, the TCA cycle, the urea cycle, fatty acid metabolism, and glycogen metabolism, is regulated by acetylation. One study reported that the expression of epigenetic modifiers of histone acetylation was significantly upregulated during aestivation in *A. japonicus*, promoting deacetylation and thereby reducing gene expression.

Reversible protein phosphorylation (RPP) is a mechanism used by many aestivation animals to regulate post-translational modifications of proteins. For instance, in the land snail *Otala lactea*, RPP regulates the activity of various enzymes involved in carbohydrate catabolism such as glycogen phosphorylase, phosphofructokinase, pyruvate kinase, pyruvate dehydrogenase, glucose-6-phosphate dehydrogenase, glutamate dehydrogenase, sodium-potassium ATPase, sarco(endo)plasmic reticulum Ca-ATPase, eIF2, and eEF2 to inhibit their function during aestivation [54]. In the sea cucumber *A. japonicus*, aestivation induces changes in the phosphorylation status of six major functional proteins, including those involved in protein synthesis (eIF2, eIF4, eEF2), transcriptional regulation (SIN3B, NF-ƙB1/p105), kinases (CaMKIId, GSK-3β, RSK, aPKC, PKC1, PKC2), signal transduction (TBC1D4, MAPK, ERK), transporter proteins (CACNA1A, GLUT5), and DNA binding factors (Histone 3.3). These changes are involved in regulating various biological processes such as protein translation inhibition, high-temperature cytoprotection, cell cycle and apoptosis regulation, and glycogen synthesis [55].

##### Changes in miRNA Expression

MiRNAs are essential regulators of post-transcriptional gene expression, playing roles in mRNA transcript storage, degradation, and translation. The integration of miRNAs into the miRNA-induced silencing complex (miRISC) leads to the recruitment of the RNA-induced silencing complex (RISC) to target mRNA, resulting in the reversible repression of translation through mRNA degradation, a process facilitated by AGO2 (argonate endonuclease) within the complex. Several miRNAs that regulate transcription during aestivation have been identified, particularly in the sea cucumber *A. japonicus*. For instance, in the intestine of aestivating animals, miR-22 is overexpressed, and it operates with the AKT kinase and PTEN phosphatase in the PTEN-AKT signaling pathway to respond to extracellular signals that inhibit the AKT pathway. The AKT pathway is a major regulatory pathway in cellular anabolism, promoting protein synthesis and cell cycle processes, while PTEN inhibits Akt signaling by dephosphorylating components of this pathway [51,56]. Additionally, in the intestine of aestivating sea cucumbers, miR-200-3p is overexpressed. This miRNA targets a gene encoding enoyl coenzyme A hydratase and 3-hydroxyacyl coenzyme A dehydrogenase (EHHADH), which are essential components of the peroxisomal fatty acid β-oxidation pathway, suggesting that miR-200-3p plays a role in regulating fatty acid metabolism during aestivation [56]. The interaction between miR-200-3p and AjCA was confirmed recently, which suggests that miR-200-3p may temporarily block cell cycle progression during aestivation by repressing cyclin A mRNA translation [57]. Additionally, miR-9, miR-10a, miR-92, and miR-124 overexpression may regulate apoptosis during aestivation [56]. MiR-7 was significantly upregulated in the intestine of aestivating sea cucumbers and may play a role in regulating intestinal epithelial cell differentiation. Recently, studies have discovered several novel miRNAs in the sea cucumber *A. japonicus*. Five of the top six most highly expressed sea cucumber-specific miRNAs are related to their adaptive evolutionary traits, including aestivation and regeneration. The predicted targets of these miRNAs are involved in pathways related to signal transduction, metabolic regulation, organ development, and apoptosis regulation that are associated with the JAK-STAT and Wnt signaling pathways. These pathways are known to play key roles in aestivation and intestinal regeneration in animals, including sea cucumbers [58].

##### Isolated Preservation of mRNA Transcripts

During hypometabolism, many transcripts are expressed but not translated and are preserved in isolation until awakening to be used for rapid protein synthesis. This isolation is facilitated by RNA-binding proteins, such as TIA-1 (T-cell intracellular antigen-1), TIAR (T-cell inhibitor of apoptosis-related protein), and PABP-1 (poly A-binding protein-1), that are up-regulated during hypometabolism. These RNA-binding proteins preserve mRNA by affecting transcription, mRNA splicing, or post-transcriptional translation [51].

#### 1.2.8. The Regulatory Network of Aestivation

In summary, aestivating animals respond to high environmental temperatures, prolonged immobility, food shortages, and reduced metabolism via various physiological and biochemical strategies. These strategies include antioxidant defenses, nitrogen metabolism and ammonia detoxification, lipid storage and metabolism, anti-degenerative muscle atrophy, antibacterial immune protection, and gut degeneration and regeneration. Each of these strategies has its own regulatory mechanisms, and they also interact and coordinate with one another to form an integrated regulatory network. Based on previous studies of aestivation (see Table 1), this paper outlines some of the regulatory networks and metabolic processes that have been demonstrated to support aestivation.

## 2. Characteristics and Research Progress of Vertebrate Aestivation

### 2.1. Amphibians—African Clawed Frog (Xenopus laevis)

The African clawed frog (*Xenopus laevis*) is an aquatic amphibian native to sub-Saharan Africa. During periods of seasonal drought, the water in which these frogs reside often dries up, prompting them to migrate to a new water habitat or enter a dormant state known as aestivation (Figure 1a). During aestivation, African clawed frogs experience severe dehydration but can tolerate up to 35% of the total body water lost [61]. As a result, the osmolality of their body water increases, which helps prevent further water loss. One of the substances used to increase osmolality is urea. Plasma and liver urea concentrations can increase 15–30-fold and ~20-fold, respectively, during dehydration. Severe dehydration leads to an increase in blood viscosity and a decrease in blood volume, both of which impair oxygen transport and cause hypoxic stress. As needed, African clawed can shift from aerobic to anaerobic metabolism during aestivation to meet their energy requirements under hypoxic conditions [62]. Additionally, inactivity during aestivation can lead to severe muscle atrophy in non-adapted animals due to months spent in burrows during aestivation. Liver and intestinal cells may atrophy due to prolonged non-feeding [63,64]. Therefore, mechanisms that promote cell survival and inhibit apoptotic responses are particularly important.

Research on aestivation in African clawed frogs has focused mainly on the effects of dehydration on metabolic and oxidative stress processes. Recently, various pathways regulating the oxidative stress response in these frogs have been studied. For instance, Luu et al. examined the role and regulation of the Akt-FoxO signaling pathway in the skeletal muscle of African clawed frogs, demonstrating that Akt inhibits the activity of FoxO1 and FoxO3 via protein phosphorylation. On the other hand, inhibition of Akt under dehydration stress may activate FoxO transcription factors [34]. Meanwhile, Zhang et al. found that FoxO4 binding to DNA increased during dehydration, and the decrease in FoxO4 phosphorylation levels at Ser-193 mediated by Akt likely led to increased FoxO4 activity [65]. In addition, Wu et al. studied the insulin signaling cascade in *X. laevis* under dehydration stress and reported reduced phosphorylation levels of the insulin growth factor-1 receptor (IGF-1R) in the brain and heart, followed by decreased phosphorylation levels of Akt and mTOR kinase regulatory targets, indicating the importance of this pathway under dehydration stress [66]. Wu et al. examined the regulation patterns of key mitogen-activated protein kinases (MAPK) (including JNK and MSK), its downstream transcription factors (c-Jun and ATF2), and several heat shock proteins (HSP60, etc.) in the brain and heart of African clawed frogs under dehydration stress. The study aimed to understand the role of the MAPK cascade in dehydration based on previous findings about microRNA action in African clawed frogs. The results suggest that specific patterns of inhibition of MAPK, downstream transcription factors, and HSPs under dehydration stress may contribute to establishing a survival-promoting state [67]. In addition to the MAPK cascade, other studies have focused on different metabolic pathways during aestivation in *X. laevis*. For example, glycolysis is a key metabolic pathway under hypoxia. Katzenback et al. investigated the regulation of lactate dehydrogenase (LDH), a key anaerobic metabolic enzyme under dehydration stress in African clawed frogs, and found that LDH function was regulated by post-translational modifications together with urea levels [68]. Moreover, Dawson et al. investigated transcript levels and phosphorylation status of a key glycolytic enzyme, pyruvate kinase (PK), in the liver and skeletal muscle of *X. laevis* under dehydration stress and found that dehydration-induced phosphorylation modifications of PK may be crucial in regulating the preservation of PK function [69].

Additionally, research on African clawed frogs also focused on muscle and visceral atrophy during aestivation. Biggar et al. conducted a study on miRNAs in aestivating African clawed frogs, which showed that pro-apoptotic miRNAs can inhibit cell proliferation by causing cell cycle arrest, whereas anti-apoptotic miRNAs may help prevent skeletal muscle atrophy during aestivation [70]. Furthermore, Tamaoki et al. investigated the morphological, biochemical, transcriptional, and epigenetic changes in the intestine of African clawed frogs during fasting and re-feeding periods of aestivation. Their findings indicate that the intestine undergoes overall metabolic repression at the transcriptional level to conserve energy during fasting, but this state is rapidly reversed upon refeeding and is regulated via histone acetylation, methylation, and phosphorylation modifications of RNA polymerase II [71].

### 2.2. Fish—African lungfish (P. annectens)

Possessing lungs can enable some species to survive without food and water for long periods of time during aestivation due to their ability to extract maximal energy (ATP) from aerobic catabolism of stored fuels [72]. There are six extant species of lungfish: one in Australia (*Neoceratodus forsteri*), one in South America (*Lepidosiren paradoxa*), and four in Africa (*Protopterus aethiopicus, P. annectens, P. amphibius, and P. dolloi*) [73,74]. Although Australian lungfish cannot aestivate and South American lungfish can aestivate in mud for 3–4 months during seasonal drought without forming cocoons, African lungfishes are the most adept at long-term survival since they can aestivate in dry mucus cocoons for up to five years [75,76,77]. Here we will concentrate on aestivation in *P. annectens*, the most widely studied lungfish (Figure 1b). This species aestivates in air or mud wrapped in a completely dry mucus cocoon and tolerates prolonged periods of water and food deprivation (months or years) in high-temperature environments [16,78,79]. However, as soon as aestivating lungfish are exposed to water, they awaken and emerge from their cocoons [77]. Aestivation is characterized by metabolic depression, including reduced pulmonary respiratory rate, reduced oxygen consumption, and reduced cardiovascular activity (low heartbeat, blood pressure, and blood flow [78]. To maintain a low rate of waste production and avoid excessive ammonia toxicity, aestivating lungfish can mobilize stored proteins as fuels but convert waste ammonia into urea, which raises body fluid osmolality and contributes to water conservation. Moreover, cell death and tissue degradation are prevented during aestivation, particularly in organs where morphofunctional integrity is vital for survival. For instance, the skeletal muscles of aestivating lungfish are immobilized for long periods, but disuse atrophy of skeletal muscles can be prevented by protective mechanisms [80].

In recent years, an increasing number of studies have been dedicated to exploring the physiological, biochemical, and molecular mechanisms of African lungfish (*P. annectens*) during aestivation. One example is a study of the involvement of molecular components of the NOS/NO system in tissue-specific morphofunctional remodeling that occurs during both the maintenance and awakening periods of aestivation. Amelio and Garofalo conducted experiments employing Western blotting and immunofluorescence microscopy to examine the localization and expression of endothelial-like nitric oxide synthase (eNOS) isoforms, along with their chaperones Hsp-90 and transcription factors Akt and HIF-1α, in the cardiac and skeletal muscles of African lungfish. The results of their study suggest that key modifications of eNOS, Akt, Hsp-90, and HIF-1α play significant roles in the morphological and functional remodeling that occurs in organs such as the heart, skeletal muscle, lung, gill, and kidney during both the maintenance and awakening periods of aestivation [81,82,83]. Among the numerous molecular interactions that drive adaptation to environmental challenges, signaling by eNOS and its chaperone proteins may be a fundamental part of other convergent/divergent transduction cascades. Research on the mechanisms of ammonia detoxification and urea synthesis during aestivation in African lungfish has been a topic of interest. Loong et al. investigated changes in urea and ammonia levels in lungfish under both air and mud aestivation. They found that air-aestivation lungfish reduced ammonia production and increased urea synthesis to avoid ammonia poisoning, whereas mud-aestivation lungfish only inhibited ammonia production but did not increase urea synthesis [84]. Furthermore, Loong’s study of differentially expressed genes in the liver of lungfish during aestivation showed upregulation of mRNAs related to urea synthesis, including carbamyl phosphate synthase (Cps), argininosuccinate synthase (Ass), and glutamine synthetase (GS) [85]. Hiong and Chng et al. also found upregulation of mRNA expression of argininosuccinate synthase I and aminocarbonyl phosphate synthase III during the maintenance period of aestivation, indicating an increased ability of the ornithine-urea cycle to detoxify ammonia into urea [86,87]. Furthermore, Chng et al. found an increase in the abundance of two urea transporter proteins, Ut-a2a and Ut-a2b, in aestivating lungfish before the arousal period, presumably to prepare for efficient urea excretion when water was available [18]. In addition to the aforementioned research, protective mechanisms against muscle disuse atrophy during aestivation have also received significant attention. Ong and colleagues demonstrated that preventing the accumulation of homocysteine in muscle may be one of these mechanisms. They found that the mRNA and protein expression of bhmt1/Bhmt1 increased in the liver of lungfish, which helped regulate hepatic homocysteine levels during both the induction and maintenance periods of aestivation [88]. Further studies by the same group revealed that lungfish can inhibit the expression of the muscle growth inhibitor myostatin (mstn/Mstn) during aestivation, thus preventing degenerative atrophy of skeletal muscle [35]. Osmoregulation is crucial in maintaining water-salt balance during lungfish aestivation. Plasma arginine vasotocin (AVT) is a significant regulatory hormone. Konno et al. found that mRNA for the V2-type receptor of AVT is highly expressed in the kidney during aestivation in lungfish, exerting an osmoregulatory effect on water, sodium, and urea by the renal tubules [89]. The epithelial sodium channel (ENaC), which is a sodium-selective aldosterone-stimulated ion channel, regulates Na^+^ transport in tetrapods. Uchiyama et al. discovered a significant correlation between plasma aldosterone concentrations and ENaC mRNA expression in osmoregulatory organs such as the gills and kidneys of aestivating lungfish, indicating that aldosterone-dependent Na^+^ uptake mechanisms via ENaC may be present in the epithelial cells of the osmoregulatory organs of lungfish [90]. Furthermore, African lungfish must prevent the formation of thrombi during aestivation. Transcriptomic studies have revealed that the maintenance phase of aestivation in lungfish is characterized by reduced expression of genes associated with coagulation, complement fixation, and iron-copper metabolism, potentially preventing thrombosis [85,86]. Hiong et al. also investigated the gene expression of coagulation factor II (F2) and fibrinogen gamma chain (Fgg) and showed that lungfish avoid thrombosis by reducing mRNA expression levels of f2 and fgg during the maintenance phase of aestivation [91].

### 2.3. Reptiles—Turtles and Crocodiles

Aestivation has been reported in reptiles, with summer aestivation by some turtle species being the most frequently studied. For instance, long-neck turtles (*Chelodina rugosa*) bury themselves in mud in the late dry season every year and undergo aestivation for 4–5 months (Figure 1c). After 2 weeks of aestivation, their metabolic rate drops to 28% of the standard metabolic rate (SMR), a process that conserves energy and water to prolong their survival in dry conditions [92]. Another freshwater turtle species (*Chelodina longicollis*) weighs their energy and water availability as their habitat dries out to determine whether to go into aestivation or migrate to a new habitat [93,94]. Egyptian tortoises (*Testudo kleinmanni*) typically enter aestivation between mid-May and early June each year, and they use rodent burrows, crevices, or vegetation as a refuge for aestivation due to their small size [95,96,97]. Western swamp turtles (*Pseudemydura umbrina*) aestivate underground for six months each year during the dry season of the swamp [98], whereas yellow mud turtles (*Kinosternon flavescens*) aestivate in sand dunes in June and July each year [99]. Aestivation has also been observed in at least 10 crocodilian species. Freshwater crocodiles (*Crocodylus johnstoni*), for instance, undergo aestivation underground for 3–4 months each year during the dry season [100,101]. However, research on aestivation in reptiles has mostly focused on physiological and metabolic changes and energetic coordination, with relatively little research conducted on the biochemical and molecular aspects.

**Figure 1 ijms-24-14093-f001:**
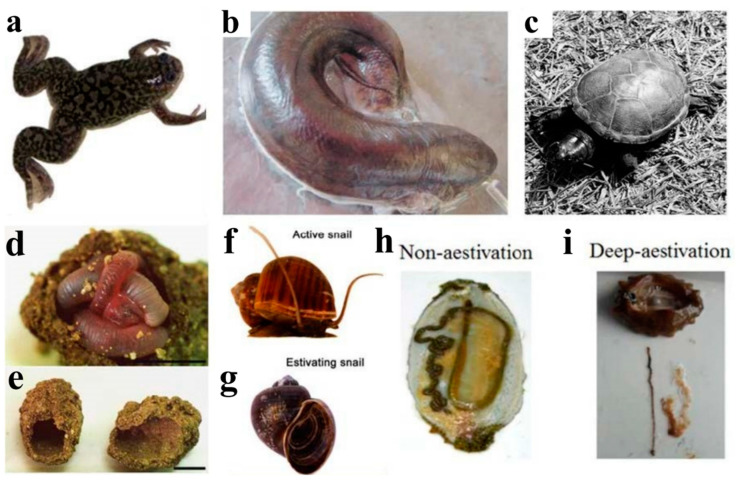
The major species studied as aestivation models. (**a**) Aestivating frog [67] (**b**) Aestivating lungfish with a mucus cocoon on the body surface [76] (**c**) Aestivating turtle [99] (**d**,**e**) Aestivating earthworm [102] (**f**,**g**) active and estivating snail [25] (**h**) Anatomy of a non-aestivating sea cucumber [55] (**i**) Anatomy of an aestivating sea cucumber (noticeable degeneration of the intestine and respiratory tree) [55].

## 3. Characteristics and Research Progress of Invertebrate Aestivation

### 3.1. Annelids—Earthworms

Changes in soil moisture content are the main environmental trigger for earthworm aestivation. When soil water content decreases during the dry season, earthworms cease feeding, empty their guts, and migrate to deeper soil layers, where they form aestivation chambers to encase themselves and minimize water loss (Figure 1d,e) [103,104,105]. Aestivation in tropical earthworms has been observed since the early 20th century and classified into different types based on various factors, such as the depth of the aestivation soil layer, the formation of aestivation chambers, and mucus spheres [106]. Some less mobile earthworms survive the dry season by creating drought-resistant cocoons on the soil surface as their main survival strategy [107,108,109]. More mobile earthworms, on the other hand, choose to survive the dry season by forming non-permanent horizontal burrows deep in the soil or by forming aestivation chambers covered with mucus and gut contents [110,111,112,113,114].

Bayley et al. conducted a further exploration into the metabolic characteristics of the earthworm *Aporrecodea caliginosa* during aestivation. They found a significant decrease in metabolic and gas exchange rates, a significant increase in urea concentrations, and an association between the formation of aestivation chambers and increased osmotic pressure of body fluids [110]. Additionally, the concentration of alanine significantly increased during aestivation, potentially acting as a nitrogen reservoir during dehydration and storing ammonium ions from other amino acids as a supplementary energy source. Holmstrup et al. obtained similar results in experiments with the earthworms *Aporrectodea tuberculata*, *A.*, and *A. longa*, hypothesizing that the increased concentration of alanine during aestivation was intended to reduce the rate of water loss from the body [115]. Alanine is also a well-known end product of anaerobic glycolysis in many invertebrates, typically linked 1:1 with succinate production [116]. Extensive research has been conducted on this phenomenon, particularly in intertidal and shallow subtidal species such as oysters, mussels, limpets, and other snails during periods of aerial exposure at low tide [117]. Considering the potential low oxygen conditions, it is plausible that certain estivating animals may also utilize this pathway. All can be reversed or oxidized when oxygen becomes available again. Tilikj et al. conducted a transcriptomic study on the aestivating earthworm *Carpetania matritensis*, finding overall metabolic and digestive repression during aestivation, significant reductions in protein turnover and macromolecular metabolism, and significant up-regulation of antioxidant genes, DNA repair genes, and immune response genes, potentially related to oxidative stress caused by excessive ROS production during aestivation [118]. Although the phenomenon of aestivation in earthworms has been reported for a long time, relatively little research has been conducted on the physiological, biochemical, and molecular aspects of earthworm aestivation, and more in-depth studies are needed to elucidate these mechanisms in the future.

### 3.2. Molluscs

#### 3.2.1. Freshwater Snails (*Pomacea canaliculate*)

*Pomacea canaliculata*, a freshwater snail, is of interest due to its invasive nature [119,120]. The snail’s ability to endure adverse conditions is a key factor contributing to its widespread invasiveness. One survival strategy is to enter aestivation during the dry season [121]. During aestivation, snails bury themselves in mud and tightly close the operculum to reduce water loss (Figure 1f,g) [10]. Like other aestivating species, the metabolic rate of these snails decreases, reducing energy expenditure but leading to a higher production of ROS during awakening [122]. Therefore, researchers are interested in studying the antioxidant defense mechanisms of aestivation in *Pomacea canaliculata*, including the role of antioxidants, protease inhibitors, chaperone proteins, and transcription factors in protecting macromolecules from oxidative stress damage during the waking stage [123]. Giraud-Billoud et al. investigated changes in TBARS, an index of oxidative stress-induced lipid peroxidative damage, in *Pomacea canaliculata*. The levels of TBARS increased significantly during aestivation but decreased to control levels upon awakening. This suggests that effective antioxidant defense mechanisms play a key role in dealing with oxidative stress caused by reoxygenation in freshwater snails [124]. Giraud-Billoud et al. reported that uric acid and reduced glutathione (GSH) may act as non-enzymatic antioxidants during arousal from aestivation, whereas antioxidant enzymes such as superoxide dismutase (SOD) and catalase (CAT) maintain constant high levels throughout the aestivation cycle, providing a permanent protective effect against ROS. Hsp70 and Hsp90 may also act as molecular chaperones during summer dormancy, protecting the proteome and providing long-term metabolic stability [13,125]. In a recent study, Giraud-Billoud et al. found that FOXO expression was upregulated during aestivation, suggesting a potential role in POS strategies [126]. Similarly, Sun et al. investigated changes in the expression of enzymes and antioxidant genes in freshwater snails during aestivation and awakening using proteomic methods. Their findings indicated that differentially expressed proteins were primarily involved in switching energy use from glucose to lipids, preventing protein degradation and improving oxidative defense, and producing purines for uric acid synthesis [25].

#### 3.2.2. Land Snails (*Theba pisana*)

*Theba pisana*, commonly known as the white garden snail, has developed several adaptations in response to heat, drought, and ion balance maintenance during aestivation [127,128]. During aestivation, they lower their metabolic rate to about 16% of the normal level and climb onto tall plants or seek shade to withstand adverse heat or drought conditions [129,130]. Moreover, they secrete a special mucus that forms a membrane to close the opening of the shell, preventing water loss [131]. Mizrahi et al. reported that *T. pisana* is the most tolerant land snail to heat and drought stress among seven Mediterranean species. This tolerance is linked to their high expression levels of heat shock protein (HSP) under non-stressed conditions, providing cytoprotection and delaying heat damage [132]. Bose et al. found that some phospholipids (e.g., LysoPC (14:0)) and amino acids (e.g., L-arginine and L-tyrosine) are upregulated in *T. pisana* during aestivation. These upregulated metabolites may play a potential role in resisting heat or drought. In addition, metabolic inhibition in land snails during aestivation may be influenced by the expression of certain central nervous system (CNS) peptides. Adamson et al. identified several peptides produced by *Theba pisana* that may be involved in the hypometabolic regulation of aestivating snails [133]. Specifically, they found that the small cardioactive peptide A (Tpi-sCAPA) was upregulated in the CNS of aestivating snails and may play a role in regulating muscle contraction in the heart (visceral ganglia) and respiration (pleural ganglia). Conversely, the abundance of the aestivation-associated peptide 12 (Tpi-AAP12) was significantly down-regulated, which may be due to its role as an antioxidant peptide regulating the significant increase in oxygen levels during the awakening stage [133]. Subsequently, Adamson et al. identified a significant upregulation of Tpi-buccalin-2 in the CNS of land snails during the awakening stage that is possibly associated with physiological responses such as muscle contraction during the transition from a hypometabolic to an active state [134].

### 3.3. Echinoderms: Sea Cucumber (Apostichopus japonicus)

Sea cucumbers, specifically *A. japonicus*, have emerged as a suitable model organism among marine invertebrates for investigating aestivation induced by environmental factors, primarily due to increased temperatures. The optimal temperature for feeding and growth of these sea cucumbers is around 14–15 °C, but when the temperature exceeds 24.5 °C, *A. japonicus* initiates aestivation (Figure 1h,i). Over the past two decades, research has focused on various aspects of aestivation in this species, including physiology, metabolism, gene expression, and epigenetic regulation.

In terms of the physiological and metabolic characteristics of *A. japonicus* during aestivation, Yang et al. studied their oxygen consumption rate (OCR) and ammonia excretion rate (AER) under laboratory conditions, demonstrating that mature sea cucumber individuals have a peak OCR and AER at 20 °C, which subsequently decrease after entering aestivation [135]. Xiang et al. quantitatively analyzed the activity and metabolites of two rate-limiting enzymes, hexokinase (Aj-HK) and pyruvate kinase (Aj-PK), to verify the inhibition of the glycolytic pathway in the intestine of aestivating *A. japonicus* and showed that transcriptional and post-transcriptional modifications might play a role in regulating glycolytic inhibition [136]. Furthermore, Xu et al. identified the distribution of fatty acids (FAs) in aestivating *A. japonicus* and discovered that FAs may be synthesized from heterotrophic bacteria during early aestivation stages, and the sea cucumber may digest long-chain FAs such as eicosapentaenoic acid (EPA) and docosahexaenoic acid (DHA) from intestinal degradation during deep aestivation stages [137]. Lastly, Wang et al. investigated enzyme activities related to immune and antioxidant defenses during aestivation, with significantly elevated activities of superoxide dismutase (SOD), catalase (CAT), and myeloperoxidase (MPO), representing enhanced immune and antioxidant defenses during aestivation [138,139].

In terms of gene expression, Xu et al. screened the internal reference genes of *A. japonicus* to improve the accuracy of the qRT-PCR techniques and found the best combination of reference genes for the intestine, respiratory tree, and muscle tissues of aestivating sea cucumbers [140]. Zhao et al. analyzed the gene expression profile of respiratory tree tissues during aestivation in *A. japonicus* and found significant upregulation of angiopoietin-like protein and KLF2/4 genes involved in antioxidant defense, as well as the bispecific phosphatase 1 gene that may be associated with the toll-like receptor signaling pathway, which plays a crucial role in the innate immune system of invertebrates against invading pathogens [141]. Chen et al. studied the overall proteomic expression profile during aestivation in *A. japonicus* and showed that carbohydrate metabolism is generally attenuated during the deep aestivation stage. They also found that proteins and phospholipids may be the main fuel sources during hypometabolism, and differentially expressed proteins in aestivation are mainly associated with protein synthesis, protein folding, DNA binding, apoptosis, cellular transport, and signal transduction, as well as cytoskeletal protein changes [142]. Xu et al. studied the expression of the apoptosis genes Ajcaspase-8 (CASP8) and Ajcaspase-3 (CASP3) and showed that apoptosis in the intestinal cells of *A. japonicus* occurred shortly after the temperature increase and most apoptotic events were completed within 20 days [45]. Gao et al. conducted a study on the spatiotemporal expression of the heat shock protein HSP70/110 in *A. japonicus* during aestivation. The results showed that AjHSP70IVAs were significantly upregulated in both body walls and muscle tissues throughout the aestivation period [143]. In a subsequent genome-wide analysis, Gao et al. found that small HSP (SHSP) proteins play a key role in stress relief in mature individuals, whereas AjHSP60/10s and AjHSP90s are more likely to play a role in housekeeping [144]. Another study by Gao et al. demonstrated that the co-chaperone protein of HSP70, DNAJ, promotes protein folding and degradation under environmental stress and that AjHSP70IVAs in *A. japonicus* are regulated by AjDNAJs during aestivation. [145]. Wang et al. discovered a potential relationship between heat shock protein 70 (HSP70) and apoptosis-inducing factor mitochondrial 1 (AIFM1) that is involved in anti-apoptotic regulation of cells during heat stress in aestivating *A. japonicus* [49]. Li et al. assembled a reference genome of *A. japonicus* and performed large-scale transcriptome sequencing to identify the key regulators and transcriptional networks involved in aestivation. They discovered that the transcription factors Klf2 and Egr1 may be associated with the regulation of the biological rhythm of aestivation and that aestivation in *A. japonicus* is regulated by the expression of the clock gene Cry1 and the transcriptional activator Bmal1. Additionally, AM7, a key module of the aestivation gene co-expression network, is involved in complex molecular regulation of multiple gene pathways, including Notch, Jak-STAT, and PI3K-Akt signaling pathways associated with cell proliferation, differentiation, and apoptosis; thyroid hormone signaling pathways associated with seasonal rhythms; neurotrophin signaling pathways associated with neuronal survival, growth, and differentiation; and Toll-like receptor signaling pathways associated with immune response [50].

In terms of epigenetic regulation, Wang et al. evaluated the role of epigenetic modifications on global gene silencing during hypometabolism in *A. japonicus*. Their results showed that DNA methylation, chromatin remodeling, histone acetylation, and histone methylation modifiers were significantly upregulated and may be involved in regulating overall transcriptional repression during aestivation. Zhao et al. found that DNA methylation may be involved in regulating transcriptional silencing and that the intestine tissue is the primary site of epigenetic regulation during aestivation. Moreover, Zhu et al. showed a positive correlation between the transcriptional expression of cyclin B and the methylation level of the core promoter, suggesting that DNA methylation is involved in regulating cyclin B transcription [146]. Yang et al. reported that differentially methylated genes (DMRs) involved in mRNA surveillance, metabolic, and RNA transport pathways, as well as 24 highly methylated Retrovirus-related Pol polyprotein from transposon (RPPT) genes and 15 hypomethylated genes, were identified in *A. japonicus* during aestivation [53]. Chen et al. investigated reversible protein phosphorylation in aestivating *A. japonicus* and observed hyperphosphorylation in functional proteins involved in protein synthesis, transcriptional regulation, kinases, signal transduction, translocation, and DNA binding [55]. Chen et al. analyzed the miRNA expression profiles in different tissues of aestivating *A. japonicus* and identified several miRNAs, including miR-22, miR-92, and miR-7, that were significantly upregulated in intestine tissues and may play a role in hypometabolic regulation, apoptosis inhibition, and intestinal epithelial cell differentiation, while miR-124 and miR-9 were significantly upregulated in respiratory tree tissues and may be associated with neuronal apoptosis. Furthermore, Wang et al. found a negative correlation between miR-200-3p and AjCA (cell cycle protein A of *A. japonicus*) at the transcriptional and translational levels in intestine tissues of aestivating *A. japonicus*, suggesting that miR-200-3p may reversibly block cell cycle progression by inhibiting the translation of cyclin A mRNA transcripts [57]. Additionally, Chen et al. showed that miR-200-3p may regulate fatty acid metabolism by inducing the degradation of AjEHHADH (peroxisomal bifunctional enzyme) mRNA in the intestine tissues of aestivating *A. japonicus*.

## 4. Conclusions

Animal aestivation has been extensively studied over the years, with researchers focusing on several similar aspects across different species, providing a reference and comparative possibility for future research. In recent years, the development of high-throughput sequencing technology has greatly advanced the molecular-level study of aestivation. This article presents the latest findings and reviews the progress of animal aestivation research from the perspective of different research contents and species, aiming to reveal the physiological metabolic strategies, reversible control of gene expression, and complex signal transduction pathways and networks of aestivation and to serve as a reference for future research.

### Perspectives

(1) Aestivation is often induced by environmental factors and can be broadly classified into temperature-induced aestivation (e.g., sea cucumbers) and desiccation-induced aestivation (e.g., lungfish). Although both are responses to unfavorable environmental pressures, the mechanisms underlying the induction of aestivation may differ between these two types of stimuli. However, to date, there have been no comparative analyses focusing on the inducing factors.

(2) While the induction of aestivation is typically environmentally driven, the potential for aestivation is highly conserved among species. For example, non-aestivating species often struggle to evolve the behavioral traits associated with aestivation, even under extreme temperature and drought stress. Current research on aestivation has mainly focused on individual aestivating species, lacking comparative analyses between species or different populations within the same species. Future studies should employ comparative genomics, functional genomics, and epigenetics to explore the adaptive evolutionary patterns of aestivation in different species facing diverse environmental pressures.

(3) Aestivation is an adaptation of organisms to seasonal changes in the natural environment, and the behavioral and physiological activities of aestivating organisms exhibit periodic changes corresponding to the seasons. Therefore, it is likely that aestivation is influenced by biological clocks. Currently, only a few studies have examined the expression patterns of clock genes during different stages of aestivation, and the relationship between environmental stress perception, circadian rhythm regulation, and aestivation remains largely unexplored.

(4) Most current research has focused on comparing physiological responses between the non-aestivation period and the aestivation period, resulting in a better understanding of the physiological characteristics of organisms during the low metabolic state of aestivation. However, little attention has been paid to the processes of aestivation induction and arousal. In these processes, researchers can search for temperature/desiccation sensors in organisms and investigate the signal transduction and regulatory mechanisms between these sensors and the switch controlling aestivation. These areas of study hold great interest and potential.

## Figures and Tables

**Table 1 ijms-24-14093-t001:** The regulatory network of aestivation.

Strategy	Pattern Diagram	Ref
1. Anti-oxidant defense	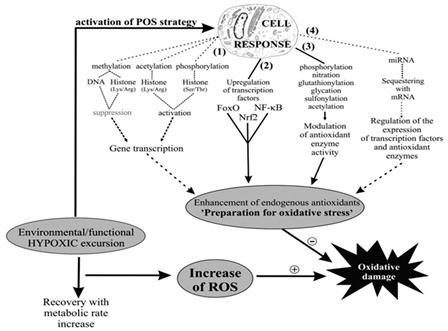	[9]
2. Nitrogen metabolism and ammonia detoxification	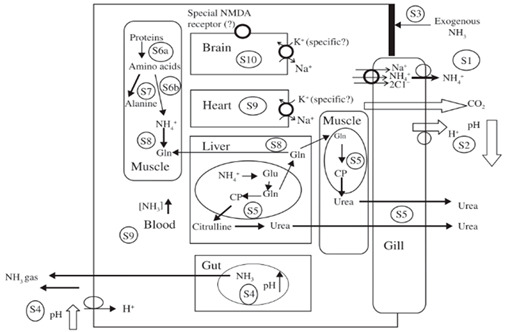	[59]
3. Lipid reserves and lipid metabolism	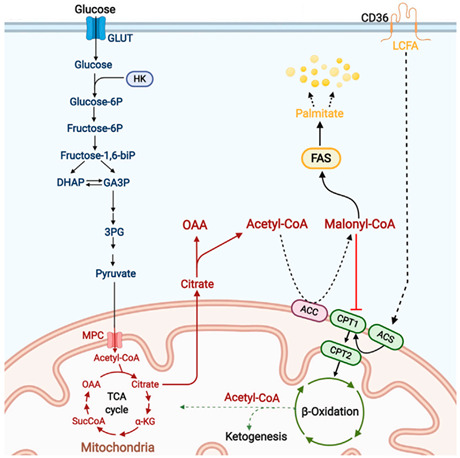	[26]
4. Muscle disuse atrophy and muscle protection	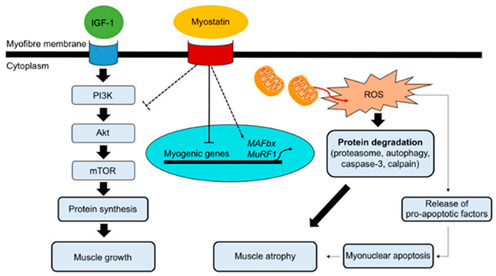	[60]
5. Antibacterial immune protection	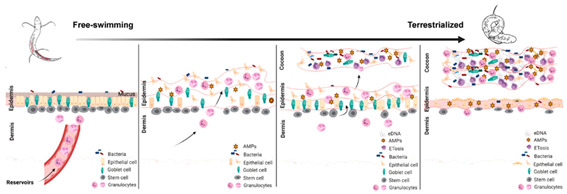	[2]
6. Visceral degeneration and regeneration.	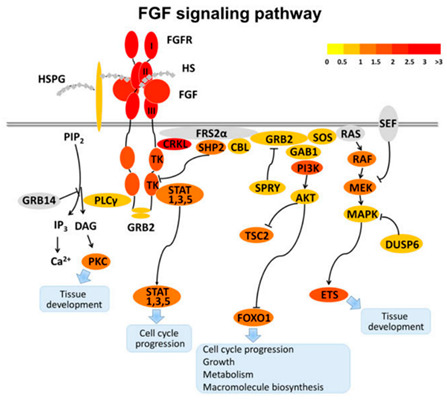	[50]

## Data Availability

Not applicable.

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
