# Peer review of "Aestivation in Nature: Physiological Strategies and Evolutionary Adaptations in Hypometabolic States"

_ijms, 2023, doi:10.3390/ijms241814093_

Round 1

Reviewer 1 Report

The authors showed the adaptation of organisms to seasonal changes by showing some metabolic and physiological solutions of summer organisms, with a broad overview of the species.

The types of adaptation and the various conditions to which to adapt have a very broad vocabulary which is not easy to metabolise for a non-specialist reader in the sector. I would suggest adding an outline to the text in which to define the different adaptive strategies which could also be useful as a glossary.

I appreciated the presence of metabolic patterns, but the manuscript is still descriptive because it lacks the genomic part related to hypometabolic states and aestivation, which could explain many details. In recent years, many articles have been published about these states. With the great development of omics disciplines, some answers can already be sought in these technologies and not in the future.

I suggest adding a paragraph explaining at least a rough outline of the use of the homics approach and its utility in this field.

Author Response

Thank you for your valuable feedback on the manuscript. I agree that the vocabulary used to describe the types of adaptation and the conditions they adapt to can be challenging for non-specialist readers. To address this, I have included an outline in the text that defines the different adaptive strategies to aid readers in understanding the concepts.

The rapid development of omics has indeed helped us gain a deeper understanding of the potential induction and maintenance mechanisms of aestivation at the molecular level. This review mainly focuses on the information provided by transcriptomics, metabolomics, proteomics, and epigenomics-related articles. Many results from omics are mentioned throughout the manuscript, such as in section 1.2. Currently, our understanding of the potential regulatory pathways of aestivation is mainly based on the development and application of omics technologies. However, at the genomic level, there is indeed limited research that reveals the underlying mechanisms of aestivation. From my understanding, there have been many articles on genomics methods in the study of mammalian hibernation (listed below), some of which compare the genomes of different hibernation species and obtain corresponding results, while others use resequencing techniques to genotype different populations of the same hibernation species. However, since this review only focuses on aestivation, there is not much introduction of hibernation-related research results. Compared to hibernation, the research progress of aestivation has been relatively slow. Only one article on the genome of sea cucumbers related to aestivation at the genomic level was published in 2018, and most of the genomic data of aestivating species serve as the basis for other omics analyses.

  • Bai, L., Liu, B. N., Ji, C. M., Zhao, S. H., Liu, S. Y., Wang, R., 2019. Hypoxic and Cold Adaptation Insights from the Himalayan Marmot Genome. Iscience, 11, 519-+.
  • Grabek, K. R., Cooke, T. F., Epperson, L. E., Spees, K. K., Cabral, G. F., Sutton, S. C., 2019. Genetic variation drives seasonal onset of hibernation in the 13-lined ground squirrel. Commun. Biol., 2(1), Article 478.
  • Sullivan, I. R., Adams, D. M., Greville, L. J. S., Faure, P. A., Wilkinson, G. S., 2022. Big brown bats experience slower epigenetic ageing during hibernation. Proceedings of the Royal Society B-Biological Sciences, 289(1980), Article 20220635.
  • Fu, R., Gillen, A. E., Grabek, K. R., Riemondy, K. A., Epperson, L. E., Bustamante, C. D., 2021. Dynamic RNA Regulation in the Brain Underlies Physiological Plasticity in a Hibernating Mammal. Front. Physiol., 11, Article 624677.
  • Han, Y. J., Zheng, G. T., Yang, T. X., Zhang, S. Y., Dong, D., Pan, Y. H., 2015. Adaptation of peroxisome proliferator-activated receptor alpha to hibernation in bats. BMC Evol. Biol., 15, Article 88.
  • Christmas, M. J., Kaplow, I. M., Genereux, D. P., Dong, M. X., Hughes, G. M., Li, X., 2023. Evolutionary constraint and innovation across hundreds of placental mammals. Science, 380(6643), 366-+, Article eabn3943.

Thank you again for your suggestions, and I have incorporated them into the revised manuscript.

Reviewer 2 Report

Animal aestivation is a hypometabolic states of  higly variable duration that organism enter to protect themselves from enviromental stressors. Jiang C. et al. describe several aspects of aestivation in different species such as amphibians, fish, reptiles, annelids, mollusks, and echinoderms. Authors analyze in detail the behavioral, morphological, physiological, and biochemical strategies characterizing this condition useful to face unfavorable enviromental conditions.

The review also includes recent advancements on the molecular-level study of aestivation obtained through high-throughput sequencing technology, such as reversible control of gene expression, and complex signal transduction pathways and regulatory networks that have been demonstrated to support estivation.

The topic is interesting and is covered comprehensively, in a very clear and understandable way. The review cites recent and relevant publications and includes recent findings on molecular aspects.

The manuscript should be improved by addressing a couple of points and is accepted after minor revision.

Main concerns/comments:

1.    The aspects considered are numerous, but the review is quite long. Authors should try to be as succinct as possible. For example, in some paragraphs the methodological details of the cited results are described. It is not necessary for this type of information to be part of a review. It is sufficient to cite the works listed in the bibliography. For example see Sections 1.2.6, 1.2.7.3 (lines 427-430).

2.    Abstract— I would suggest including a graphical abstract that comprehensively summarizes all types of information addressed in the test.

3.    Conclusions— The review takes into consideration the aestivation strategies in different species. As also underlined in the title, the “evolutionary” aspect should be emphasized more precisely. In the Conclusion Section, for example, it would be really interesting to underline (where possible) similarities and differences in the strategies adopted in aestivation, relating them to the evolutionary scale. This would represent a significant added value for the review.

Minor/Specific concerns:

1.    Table 1— The formatting of the table needs to be changed. Pattern diagram are completely illegible (especially in the printed version) and therefore useless. Texts fonts are too small and the figures should also be enlarged. The Ref column could be reduced by inserting only the reference number […], or by eliminating the title of the cited article which is superfluous.

2.    Please check inaccuracies in text formatting: spaces (as in line73, 157, 462, 714, etc.) and check the correct spelling of the word aestivation (sometimes spelled as estivation, as in line 457, 472,Table 1.5, etc.).

3.    Line 352— LF height, please specify the meaning of the acronym.

Author Response

Thank you for your feedback on the manuscript. I appreciate your main concerns and comments, and I will address them accordingly:

  1. I understand the review is quite long, and I agree that we should strive to be more succinct. I have specifically revised the sections 1.2.6 and 1.2.7.3 (lines 427-430) to remove unnecessary methodological details to ensure that they are more concise.

  1. I appreciate your suggestion and have included a graphical abstract that comprehensively summarizes all the types of information addressed in the text to enhance the clarity and visual representation of the review.

  1. I acknowledge the need to emphasize the evolutionary aspect more precisely, especially in the Conclusion section. I have made my best to highlight the similarities and differences in the aestivation strategies adopted by different species, relating them to the evolutionary scale in the revised version of manuscript.

I appreciate your minor concerns and comments, and I will address them accordingly:

  1. I have revised the formatting of Table 1 to ensure that the pattern diagrams are legible.

  1. I have carefully checked for inaccuracies in text formatting, including spaces (as in line 73, 157, 462, 714, etc.). Thank you! Aestivation and estivation are two different spellings for the same term, with aestivation being the British English spelling and estivation being the American English spelling. I have ensured that the term used throughout the text is consistently "aestivation."

  1. I apologize for the oversight in line 352. LF height stands for longitudinal fold height. I have made the necessary clarification in the manuscript to provide the meaning of the acronym.

Thank you again for bringing these concerns to my attention, and I have incorporated them into the revised manuscript.

Round 2

Reviewer 1 Report

I understood the reasons of the authors and appreciated the changes. I believe the manuscript is now publishable.